# Biochemical and Transcriptome Analyses Reveal a Stronger Capacity for Photosynthate Accumulation in Low-Tillering Rice Varieties

**DOI:** 10.3390/ijms25031648

**Published:** 2024-01-29

**Authors:** Mingqiang Zhu, Shan Jiang, Jinqiu Huang, Zhihui Li, Shuang Xu, Shaojia Liu, Yonggang He, Zhihong Zhang

**Affiliations:** 1State Key Laboratory of Hybrid Rice, College of Life Sciences, Wuhan University, Wuhan 430072, China; zmqjukey@whu.edu.cn (M.Z.); 2020202040061@whu.edu.cn (S.J.); 2019102040038@whu.edu.cn (J.H.); zhihui18@whu.edu.cn (Z.L.); 2020202040067@whu.edu.cn (S.X.); jsxabc@163.com (S.L.); 2Institute of Food Crops, Hubei Academy of Agricultural Sciences, Wuhan 430070, China

**Keywords:** rice (*Oryza sativa* L.), tillering, carbon and nitrogen metabolism, transcriptome, DEGs, WGCNA

## Abstract

Moderate control of rice tillering and the development of rice varieties with large panicles are important topics for future high-yield rice breeding. Herein, we found that low-tillering rice varieties stopped tillering earlier and had a larger leaf area of the sixth leaf. Notably, at 28 days after sowing, the rice seedlings of the low-tillering group had an average single-culm above-ground biomass of 0.84 g, significantly higher than that of the multi-tillering group by 56.26%, and their NSC (non-structural carbohydrate) and starch contents in sheaths were increased by 43.34% and 97.75%, respectively. These results indicated that the low-tillering group of rice varieties had a stronger ability to store photosynthetic products in the form of starch in their sheaths, which was thus more beneficial for their large panicle development. The results of carbon and nitrogen metabolism analyses showed that the low-tillering group had a relatively strong carbon metabolism activity, which was more favorable for the accumulation of photosynthesis products and the following development of large panicles, while the multi-tillering group showed relatively strong nitrogen metabolism activity, which was more beneficial for the development and formation of new organs, such as tillers. Accordingly, in the low-tillering rice varieties, the up-regulated genes were enriched in the pathways mainly related to the synthesis of carbohydrates, while the down-regulated genes were mainly enriched in the nitrogen metabolism pathways. This study provides new insights into the mechanism of rice tillering regulation and promotes the development of new varieties with ideal plant types.

## 1. Introduction

Rice (*Oryza sativa* L.) is one of the three major food crops in the world and provides staple foods for nearly half of the world’s population [1]. The yield components of rice are the number of panicles per unit area, number of spikelets per panicle, seed-setting rate, and grain weight [2,3]. The yield of rice may be increased by improving the four yield components individually or collectively [4]. In rice production practices, high yield can be achieved by two types of rice varieties. One is the multi-panicle type which mainly obtains more panicles and population spikelets by increasing the number of tillering panicles, thereby increasing the yield [5,6,7]. For these rice varieties, a large amount of tillering fertilizer is usually applied in agriculture production to obtain more rice tillers [8], and the utilization rate of this tillering fertilizer is very low, about 9–22%, resulting in a large amount of nitrogen fertilizer waste [9]. The second is the large-panicle type with an ideal plant type as its basic characteristics, increasing its number of spikelets per panicle to expand the capacity of the single-panicle sink and improve dry matter accumulation ability after heading, usually showing higher yield potential and greater nitrogen use efficiency (NUE) [10,11,12,13]. In view of this, moderate control of rice tillering and the promotion of large panicle development are important topics for future rice breeding and cultivation.

The metabolism of carbon- and nitrogen-containing compounds is fundamental to all forms of life [14]. In plants, the assimilation of inorganic nitrogen requires carbon metabolism to provide reducing equivalents, ATP and carbon skeletons [15]. The photosynthetic carbon metabolism requires nitrogen metabolism to provide the large amount of nitrogen invested in the photosynthetic apparatus, particularly rubisco and light-harvesting complexes to produce sugars, starch, organic acids and amino acids—the basic building blocks of biomass accumulation [16,17]. Plant growth and development and crop yield are highly dependent on the interaction between carbon and nitrogen metabolism [15]. Both the rice tillering stage and young panicle differentiation stage are two of the most important stages that determine the numbers of panicles and spikelets per panicle, respectively [18]—the growth and development of rice plants at these two stages are regulated by carbon and nitrogen metabolism.

Nitrogen is a primary nutrient element for plants that plays a crucial role in determining plant growth and productivity [19]. The process of nitrogen metabolism in rice mainly includes nitrogen uptake, translocation, assimilation and reuse [20]. Inorganic nitrogen is catalyzed by nitrate reductase and nitrite reductase to produce NH_4_^+^, which is converted to glutamine (Gln) and glutamic acid (Glu) through the GS/GOGAT cycle in the presence of an adequate supply of α-ketoglutaric acid (2-OG) and further converted to nitrogenous organic matter such as nucleic acids, hormones, enzymes and proteins [21]. Tiller formation depends largely on the nitrogen utilization [22]. During the vegetative growth period, the external application of nitrogen can increase the soluble protein content in rice plants and promote both the formation and outgrowth of axillary buds [23,24]. Furthermore, the activities of nitrogen assimilation-related enzymes such as NADH-GOGAT and NADH-GDH are significantly positively correlated with the number of tillers [25].

Non-structural carbohydrates (NSCs) mainly include starch and soluble sugar, which provide energy for plant growth and development [26]. In a previous study, the higher soluble sugar content in the stem or tillering nodes was more conducive to development and improving the survival rate of tillers, thereby increasing the spike rate [27,28]. Sucrose is the main form of carbohydrate transport in plants [29]. The exogenous application of sucrose can relieve the dormancy of axillary buds and promote axillary bud growth [30]. Starch is one of the main long-term storage substances in plants [31]. During the vegetative and early reproductive phases of cereal development, starch is stored in vegetative sink tissues such as the stem and leaf sheaths, converted into sucrose, and transported to the reproductive sink tissue, the filling grain, during the later growth and development stages [32].

In this study, two groups of rice varieties with significantly different tillering characteristics were screened through field experiments. We investigated the tillering-associated traits of these two groups of rice varieties and their relationship with other agronomic traits and analyzed the effect of the balance of carbon and nitrogen metabolism on the tillering of rice. In addition, the genetic regulation mechanism between the two groups of rice varieties was studied at the transcriptome level, which provided a more scientific basis for analyzing the physiological and genetic regulation mechanism of rice tillering.

## 2. Results

### 2.1. Rice Tillering Characteristics and Related Phenotypic Traits

The tiller numbers of the tested rice varieties were investigated and a wide range of variation among the 13 varieties in this trait at 34 days after sowing was observed (Figure 1a). Variety 9311PAY1 had the lowest tiller numbers, with an average of only 1.73 tillers per plant, and Y4B had the highest tiller numbers, with an average of 6.78 tillers per plant. Based on the significance of the difference in the tiller numbers per plant among the tested varieties, the varieties could be categorized into low-tillering and multi-tillering groups; the low-tillering group included five varieties (9311PAY1, V564, R900, R2257 and CZ2H) with the tiller numbers per plant ranging from 1.73 to 2.73 and the average tiller number being 2.12. The multi-tillering group contained six varieties (Y4B, DG, GC2H, FH838, HHZ and JY948) with the tiller numbers per plant ranging from 3.78 to 6.78 and the average tiller number being 5.22, which was significantly different from that of the low-tillering group (Figure 1b). In terms of tillering dynamics, the low-tillering group started tillering at 16 days after sowing, while the multi-tillering group started at 13 days after sowing. The low-tillering group stopped tillering at 34 days after sowing, while the multi-tillering group rice varieties continued to grow tillers 40 days after sowing (Figure 1c). The plant morphology of the two groups of rice varieties was significantly different (Figure 1d). The leaf age of the low-tillering group rice varieties was extremely significantly lower than that of the multi-tillering group rice varieties at 13 days after sowing (Figure 1e) and the average leaf age of the low-tillering group rice varieties was 8.24 at 34 days after sowing, which was still significantly lower than that of the multi-tillering group rice varieties (Figure 1f).

In addition, we investigated the size of the sixth leaf for the two rice groups. The average leaf length and width of the sixth leaf in the low-tillering group were 26.17 cm and 0.92 cm, respectively, which were significantly higher than those of the multi-tillering group with 20.15 cm (Figure 1g,j) and 0.66 cm (Figure 1h,k), respectively. As a result, the average leaf area of the sixth leaf of the low-tillering group was 18.03 cm^2^, which was a significant increase of 79.34% in contrast to that of the multi-tillering group (Figure 1i,l).

### 2.2. The Accumulation and Distribution of Dry Matter in Rice Seedlings

We examined the dry matter accumulation of the rice seedlings at 28 days after sowing. It was found that the single-stem biomass for the rice seedlings of the low-tillering group was significantly greater than that of the multi-tillering group (Figure 2a), and the average single-stem biomass for the low-tillering group seedlings (0.84 g) was significantly greater than that of the multi-tillering group by 56.26% (Figure 2b), although there was no significant differences in the aboveground biomass per plant between the two groups of rice varieties (Figure 2c). Furthermore, compared to the multi-tillering group of rice varieties, the low-tillering group of rice varieties had significantly higher sheath NSC and starch contents, with increases of 43.34% (Figure 2d,h) and 97.75% (Figure 2e,i), respectively, and there were no significant differences in the sheath soluble sugar content and sucrose content between the low-tillering group and the multi-tillering group (Figure 2f,g). There were no significant differences in leaf NSC (non-structural carbohydrate) content, starch content, soluble sugar content or sucrose content between the two groups of rice varieties (Figure 2j–m). As we know, sheath NSC is an important storage form of photosynthetic products in rice seedlings, including starch, soluble sugar and sucrose, etc., among which starch is the most stable storage substance in NSC. These results indicated that the low-tillering group of rice varieties had a stronger ability to store photosynthetic products in the form of starch in their leaf sheaths.

### 2.3. Carbon and Nitrogen Metabolites and Enzyme Activities of the Rice Seedlings

Some of the nitrogen metabolism-related indicators of the two groups of rice varieties were measured. Compared to the multi-tillering group of rice varieties, the low-tillering group of rice varieties had significantly lower leaf soluble protein contents (Figure 3a,m) and glutamate (Glu) contents (Figure 3b,n), with decreases of 51.01% (Figure 3a,m) and 19.70% (Figure 3b,n), respectively. No significant differences were observed between the two rice groups in the contents of free amino acids, glutamine (Gln), α-ketoglutarate or nitrate nitrogen in leaves (Figure 3c–f). The assays on the main enzyme activities of carbon and nitrogen metabolism showed that the leaf glutamate synthase (GOGAT) activity of the low-tillering rice group was significantly lower than that of the multi-tillering rice group, with a decrease of 25.44% (Figure 3g,o), and no significant differences were observed between the two rice groups in the activities of leaf nitrogen metabolism-related enzymes, such as glutamate dehydrogenase (GDH), nitrate reductase (NR) and glutamine synthetase (GS), as well as the major leaf carbon metabolism-related enzymes, sucrose-phosphate synthase (SPS) and sucrose synthase (SS) (Figure 3h–l). The above results indicated that the low-tillering group of rice varieties had a relatively strong carbon metabolism activity that was favorable for the accumulation of photosynthesis products, while the multi-tillering group of rice varieties had relatively active nitrogen metabolism activities that were more favorable for the development and formation of new organs, such as tillers.

### 2.4. Identification and Functional Enrichment Analysis of Core Conserved DEGs of Rice Seedlings at the Tillering Stage

To understand the genetic regulation mechanism of rice tillering, the tiller nodes of four low-tillering rice varieties—9311PAY1 (L1), V564 (L2), R900 (L3) and R2257 (L4)—and three multi-tillering varieties—Y4B (M1), DG (M2) and GC2H (M3)—were collected for RNA sequencing. A total of 176.81 Gb of clean data was obtained from 21 libraries, each with a Q30 base percentage of 93.19% or higher (Appendix A). About 93.45–94.82% of clean reads could be successfully mapped on the reference genome. Furthermore, PCA (principal component analysis) showed that the samples were clearly clustered into two groups and Spearson’s correlation analysis showed a good reproducibility among the three biological replicates (Appendix A). To further validate the reliability of the RNA-seq data, seven genes were randomly selected for performing quantitative qRT-PCR in the current study. As shown in Appendix A, the qRT-PCR detected a similar expression tendency to that of the RNA-Seq data. These results indicated that the transcriptome sequencing data were reliable.

Then, the DEGs (differentially expressed genes) of the two groups of rice varieties were identified according to the criteria of |log2Fold Change| ≥ 1 and *p*-adjust < 0.05. The results showed that the DEGs of L2_vs_M2 were the least, with 2148 DEGs (1198 up-regulated genes and 950 down-regulated genes), and the DEGs of L4_vs_M1 were the most, with 7902 DEGs (3737 up-regulated genes and 4165 down-regulated genes) (Figure 4a). Further analysis of the DEGs shared by each low-tillering rice variety and the three multi-tillering rice varieties showed that the number of common DEGs in L1_vs_M, L2_vs_M, L3_vs_M and L4_vs_M was 615, 669, 1176 and 1442, respectively (Appendix A). In summary, the results showed that there was a large difference in the transcriptome level between the two groups of rice varieties, and there was also a large difference in the transcriptome level between the four low-tillering varieties.

The Venn analysis of 12 comparison pairs identified 74 common DEGs between the low-tillering and multi-tillering rice groups (Figure 4b). Of these DEGs, there were 34 genes significantly up-regulated and 40 genes significantly down-regulated in the low-tillering rice varieties (Figure 4c–e). KEGG enrichment analysis showed that the up-regulated genes in the low-tillering rice varieties were mainly enriched in the carbohydrates synthesis-related pathways, such as “flavonoid biosynthesis”, “phenylpropanoid biosynthesis”, “terpenoid backbone biosynthesis”, “stilbenoid” and “diarylheptanoid and gingerol biosynthesis” (Figure 4f). As for the examples of the DEGs up-regulated in the low-tillering group, *Os12g0271700* might encode solanesyl-diphosphate synthase 3, which is mainly involved in terpenoid backbone biosynthesis, and both *Os03g0568400* and *Os10g0196000* belong to the Cytochrome P450 family, which is mainly involved in stilbenoid, diarylheptanoid, gingerol, flavonoid and phenylpropanoid biosynthesis. By contrast, the down-regulated genes in the low-tillering rice varieties were mainly enriched in the nitrogen metabolism pathways, such as “alanine, aspartate and glutamate metabolism” and “butanoate metabolisms” (Figure 4f). For example, *Os10g0213100*, one of the genes significantly down-regulated in the low-tillering group of rice varieties, might encode aminotransferase class III, which is mainly involved in amino acid and butyric acid metabolism. These results may imply that seedlings at the tillering stage of the low-tillering rice group have relatively strong carbon metabolism and relatively weak nitrogen metabolism.

### 2.5. The Identification of DEGs Associated with Rice Tillering Regulation by WGCNA

We further analyzed the genes differentially expressed in rice seedlings at the tillering stage between the two groups of rice cultivars by weighted gene co-expression network analysis (WGCNA) and constructed a co-expression network of the genes closely related to the tillering traits investigated in the current study. After removing genes with low expression and low coefficient of variation (FPKM < 1 and CV < 0.1), a total of 37 gene co-expression modules were identified and, of these, 21 modules were significantly correlated with the tillering-associated traits (Figure 5a). Of these, two modules (MEroyalblue and MElightgreen) were most significantly correlated (negatively and positively, respectively) with the major trait tiller number per plant, with correlation coefficients of −0.738 (*p* < 0.001) and 0.734 (*p* < 0.001), respectively.

MEroyalblue was negatively correlated with the traits leaf age (LA), soluble protein (SP) and glutamate (Glu) contents, and glutamate synthase (GOGAT) activity, with correlation coefficients ranging from −0.513 to −0.877 (*p* < 0.05) (all of these traits were significantly and positively correlated with tiller number per plant), and positively correlated with the traits leaf length (LL), leaf width (LW), leaf area (L_Area), single-stem biomass (SSB), sheath NSC content (SNSC) and sheath starch content (SS), with correlation coefficients ranging from 0.712 to 0.925 (*p* < 0.001) (all of these traits were significantly and negatively correlated with tiller number per plant) (Appendix A). KEGG functional enrichment analysis showed that the genes in MEroyalblue were mainly involved in carbon-related metabolic pathways, such as “glyoxylate and dicarboxylate metabolism”, “carbon fixation in photosynthetic organisms”, “starch and sucrose metabolism”, “phenylpropanoid biosynthesis” and “zeatin biosynthesis” (Figure 5b and Appendix A). The heat map showed that the expression levels of these genes in the MEroyalblue module were significantly higher in the four low-tillering rice varieties (Figure 5c).

On the contrary, MElightgreen was significantly and positively correlated with the traits SP and Glu contents and GOGAT activity, with correlation coefficients ranging from 0.468 to 0.603 (*p* < 0.05), and negatively correlated with the traits LL, LW, L_Area, SSB, SNSC and SS contents, with correlation coefficients ranging from −0.542 to −0.777 (*p* < 0.05). KEGG functional enrichment analysis showed that the genes in MElightgreen were mainly involved in nitrogen metabolic pathways (such as “biotin metabolism”, “glutathione metabolism”, etc.) and energy metabolism pathways (“fatty acid biosynthesis”, “ABC transporters”, “nicotinate and nicotinamide metabolism”, etc.) (Figure 5b and Appendix A). The results of gene expression heat map analysis showed that the genes in MElightgreen were significantly down-regulated in the four low-tillering rice varieties (Figure 5d).

The top 30 connected genes for the MEroyalblue and MElightgreen modules were selected for co-expression network construction, respectively (Figure 5e,f). When the top 10 connected genes in the two modules were picked out by their connectivity values, respectively (Appendix A), it was found that none of the genes in each of the two modules analyzed above had been previously studied and reported in detail. Their biological functions are worthy of further study in the future.

Transcription factors (TFs) are composed of a DNA binding domain that interacts with cis-regulator elements of its target genes and a protein-to-protein interaction domain that facilitates oligomerization between TFs and other regulators [33,34]. To identify hub TFs regulating rice tillering, we predicted the TFs in the MEroyalblue and MElightgreen gene modules through the transcription factor database PlantTFDB. We identified nine hub TFs, four of which had been previously studied and reported, and the other five were novel TFs. In the MEroyalblue module, there were four different TF genes, including *Os03g0411100/OsHAP2E* (NF-YA), *Os03g0619800* (B3), *Os04g0590800* (bHLH) and *Os01g0646300/OsSLRL1* (GRAS); these TF genes were up-regulated in the four low-tillering rice varieties (Figure 5g). In the MElightgreen module, there were five different TF genes, including *Os04g0443000* (FAR1), *Os04g0112200* (E2F/DP), *Os04g0495800/OsNLP2* (Nin-like), *Os11g0152700/OsbZIP79* (bZIP) and *Os12g0567300* (MYB); these TF genes were down-regulated in the four low-tillering rice varieties (Figure 5h). The expression patterns of these nine TFs in the two types of rice varieties were completely opposite, indicating that they may play important roles in the regulation of tillering in the vegetative growth period of rice.

The results above indicated that there seem to be at least two classes of genes involved in the tillering regulation of rice seedlings. The first class of genes contained in modules such as MEroyalblue are those responsible for regulating carbon metabolism and the second class of genes in modules such as MElightgreen are those for nitrogen metabolism. In the current study, the first class of genes were up-regulated and the second were down-regulated in the four low-tillering rice varieties, which might have resulted in increased carbon metabolism and decreased nitrogen metabolism in rice seedlings of the low-tillering rice varieties.

## 3. Discussion

Optimizing crop population quality is a basic way to continuously improve crop productivity [35]. The tillers of rice seedlings develop into effective tillers and ineffective tillers [36]. Ineffective tillers compete with effective tillers for nutrients, soil and sunlight, which deteriorates population quality and ultimately affects rice yield [37,38]. Of the rice varieties currently cultivated, the stem number of the population in the early stage increases too fast, resulting in excessive stems; more tillers die in the middle stage; the stem number of the population decreases rapidly; and the panicle rate decreases [39]. Controlling the growth of ineffective tillers can optimize the population structure in the later stage of rice growth, improve the photosynthetic capacity of rice and promote the development of large panicles [7,37]. The ideal rice plant type has the characteristics of fewer tillers, fewer or no ineffective tillers, more grains per panicle and thick stems, which is the basis for constructing high-quality rice populations [40]. Cultivating rice varieties with ideal plant types can effectively regulate the occurrence of ineffective tillers [41]. In this study, the tiller number of the low-tillering rice varieties increased steadily at the tillering stage and reached the maximum tiller number early at 34 days after sowing (Figure 1c), which reduced the occurrence of ineffective tillers and prevented the population from producing excessive seedling peaks, laying a foundation for optimizing the population structure during the later stages of rice cultivation.

A prerequisite for the formation of large panicles in rice is sufficient nutrient growth and the coordinated stability of carbon and nitrogen metabolism [42]. The single-stem weight at the flowering stage is significantly positively correlated with the number of spikelets. Populations with high single-stem and sheath weights give very high yields of rice [43,44]. NSCs accumulate in the stems and sheaths before flowering and then export during the grain filling stage [45,46], and starch is the predominant component of NSCs [47]. In the present study, although there was no difference in biomass per plant between the two groups of rice varieties, the low-tillering rice varieties had single-stem biomass significantly greater than that of the multi-tillering rice varieties (Figure 2a–c), and the NSC content and starch content in the sheaths of the low-tillering rice varieties were significantly higher than those of the multi-tillering rice varieties (Figure 2d,e,h,i), which provided better nutrient growth for the development of large panicles for the low-tillering rice varieties. By contrast, the multi-tillering rice varieties showed higher soluble protein content, glutamic acid content and GOGAT enzyme activity of the leaves (Figure 3a,b,g,m–o) and their higher nitrogen metabolism activity was more conducive to the formation of tillers and the development of leaves.

The transcriptome results showed that there were significant differences in the transcriptome of the tillering nodes of the two groups of rice varieties (Figure 4a,b), which is helpful to deepen our understanding of tillering regulation in rice. Elevating the nitrogen metabolism activity could promote the outgrowth of axillary buds during the vegetative stage, and strong carbon metabolism activity is conducive to the accumulation of photosynthetic products [25,48]. Furthermore, carbon metabolites, such as flavonoids, terpenoids, phenylpropanoids, etc., are involved in the synthesis of plant cell walls and cell proliferation [49,50]. In the current study, we found that the DEGs that were more significantly highly expressed in the low-tillering rice varieties were enriched in the carbohydrate synthesis-related pathways, including “flavonoid biosynthesis”, “carbon fixation in photosynthetic organisms” and “starch and sucrose metabolism”, and the lower expressed genes in the low-tillering rice varieties were enriched in the nitrogen metabolism-related pathways, such as “alanine, aspartic acid and glutamic acid metabolism” and “glutathione metabolism”, etc. (Figure 4b and Figure 5f). These results provide a good explanation for the fact that the low-tillering group of rice varieties had stronger carbon metabolism favorable for the accumulation of photosynthesis products, while the multi-tillering group of rice varieties had relatively active nitrogen metabolism more favorable for cell division and the development and formation of new organs, such as tillers.

In addition, this study identified two gene modules most significantly associated with tillering-related traits. To our regret, among the top 10 connected genes in the two modules picked out by their connectivity values, respectively (Appendix A), none have been previously studied and reported in detail. Further biological function studies of these genes in the future may identify new genes regulating tillering in rice. TFs have been proven to participate in the regulation of plant development. In this study, nine TFs were identified, and these genes had high or low expression in the low-tillering rice varieties (Figure 5g,h). Among them, *Os03g0411100/OsHAP2E* has been reported to be associated with tillering regulation [51]. *Os01g0646300/OsSLRL1* has been identified as the repressor of the GA signaling pathway to regulate internode elongation in rice [52]. In this study, these two genes were up-regulated at the tillering stage. B3 and bHLH are two large TF families in plants that play important roles in regulating plant growth and development, flower bud differentiation, and response to various stresses [53,54]. In this study, both *Os03g0619800* (B3) and *Os04g0590800* (bHLH) were significantly up-regulated in low-tillering rice varieties. *Os04g0495800*/*OsNLP2* was identified as a nitrate-responsive transcript factor regulating the expression of *OsNRs* and the activity of NR [55]. This gene was significantly up-regulated in multi-tillering rice varieties. Moreover, the E2F/DP, MYB, FAR1 and bZIP TFs control diverse biological processes, such as differentiation, development and physiological metabolism [56,57,58,59]. Likewise, we identified a total of four genes belonging to these families, and all of them had low expression in the low-tillering rice varieties. Altogether, these observations suggest that the TFs identified in the current study may play important roles in rice tillering regulation.

## 4. Materials and Methods

### 4.1. Plant Materials and Growing Conditions

The field experiment was conducted at Ezhou Experimental Base of Wuhan University (Ezhou, China, 30°23′ N, 114°52′ E). Thirteen rice varieties were selected for this study, namely, Yue 4B (Y4B), Di Gu (DG), Gui Chao 2 Hao (GC2H), Fu Hui 838 (FH838), Huang Hua Zhan (HHZ), Jia Yu 948 (JY948), Xiang Zao Xian 45 (XZX45), Chang Zao 2 Hao (CZ2H), R600, R2257, R900, V564 and 9311PAY1. On 8 August 2022, healthy seeds were soaked in clear water at 28 °C for 48 h and then germinated in Petri dishes containing wet filter papers at 32 °C for 24 h. Subsequently, the germinated seeds were sown in 434-well seedling plates with one seed in each well and grown in natural conditions. After 10 days, the rice seedlings were transplanted to the experimental field with 2 seedlings per hole. The plant density was 8.0 cm between holes within a row and 20.0 cm between rows. The soil at the experimental site was clay loam. Before transplanting, 675 kg ha^−1^ of compound fertilizer (N:P_2_O_5_:K_2_O = 16%:16%:16%) was applied as a basal fertilizer. Furthermore, in the current study, the experiment was arranged in randomized blocks with three replications.

### 4.2. Field Traits Examination

Starting from 10 days after sowing, 10 plants with normal growth were selected and tagged to regularly investigate leaf age and tiller number. At the 6-leaf stage, the 6th leaves were selected to measure the leaf length, width and area. The leaf area was calculated according to the rice length–width coefficient method (leaf area = length × width × 0.75) [60]. Each experiment had three replications, and 10 plants were measured in each replication.

### 4.3. Measurement of Above-Ground Dry Matter

At 28 days after sowing, the number of culms in five holes for each rice variety was recorded; then, the overground part of the plants was collected and kept at 105 °C for 0.5 h. Subsequently, these samples were dried to a constant weight at 65 °C and the weight of each sample was measured. The experiment had three biological replications for each variety, and 10 plants in the five holes were tested for each replication.

### 4.4. Measurement of Physiological and Biochemical Indicators Related to Carbon and Nitrogen Metabolism

At the 6-leaf stage, the leaf and sheath samples of the 5th and 6th leaves were harvested and dried in an oven for 30 min at 105 °C to deactivate enzymes and then dried at 65 °C to a constant weight. Then, the powdered samples were used to measure the content of sucrose, starch and soluble sugar according to the method described in the previous study [61]. The NSC content was calculated as the sum of the starch and soluble sugar content. All assays were performed with three biological and three technical replications.

The fresh samples of the 5th and 6th leaves were used to determine enzyme activity, as well as the contents of soluble protein, free amino acid, NO_3_^−^-N, Glu, Gln and 2-OG (α-ketoglutaric acid). The activities of SPS (sucrose phosphate synthase) and SS (sucrose synthase) were measured as described previously [62]. The activities of NR (nitrate reductase), GS (glutamine synthase), GOGAT (glutamate synthase) and GDH (glutamate dehydrogenase) were determined according to the methods described in the previous study [63]. Soluble protein and free amino acid contents were determined using the Bradford assay [64] and the ninhydrin method [65], respectively. The NO_3_^−^-N content was measured as described in the previous study [66]. The contents of α-ketoglutaric acid (2-OG), glutamic acid (Glu) and glutamine (Gln) in the leaves were calculated according to the LC-MS method described previously [67]. All assays were performed with three biological and three technical replications.

### 4.5. RNA-Seq Assembly and Analysis

At the 6-leaf stage, the tiller nodes of four low-tillering varieties—9311PAY1 (L1), V564 (L2), R900 (L3) and R2257 (L4)—and three multi-tillering varieties—Y4B (M1), DG (M2) and GC2H (M3)—were collected and stored at −80 °C. The total RNA was extracted by the Shanghai Majorbio Bio-pharm Biotechnology Co., Ltd. (Shanghai, China). After quantification by Qubit 4.0, paired-end 150bp RNA-seq was performed using the NovaSeq 6000 sequencer. The clean reads were aligned to the reference sequences of IRGSP-1.0 (https://ftp.ensemblgenomes.ebi.ac.uk/pub/plants/release-57/fasta/oryza_sativa/dna/, accessed on 17 April 2023) by Hisat2 software (version 2.1.0) [68]. The mapped reads of each sample were assembled by StringTie [69]. FPKM (fragments per kilobase of exon per million mapped fragments) was used to quantify the levels of gene expression. DESeq2 was used to identify the differential expression genes (DEGs) based on the criteria of |log2Fold Change| ≥ 1 and *p*-adjust < 0.05 [70]. KEGG pathway analysis of the DEGs was performed using the Majorbio Cloud Platform (https://cloud.majorbio.com/page/tools/, accessed on 7 August 2023).

### 4.6. qRT-PCR Analysis and Validation

To validate the reliability of the RNA-Seq data, seven genes with different expression levels and FPKM > 1 were selected to perform the qRT-PCR experiments, with a BioRad CFX-96 system (BioRad, Hercules, CA, USA) to measure their relative expression levels. The relative expression levels were calculated using the 2^−ΔΔCT^ method, as described by Livak et al. [71]. cDNA reverse transcription was performed using the All-in-One First-Strand Synthesis MasterMix (with dsDNase) kit, and qRT-PCR experiments were conducted using the Taq SYBR^®^ Green qPCR Premix kit (ABclonal, Wuhan, China). The specific primer sequences for qRT-PCR are provided in Appendix A. The reaction system and program for qRT-PCR followed the instructions provided in the kit manual. The actin gene was used as the reference (internal control) and three technical replicates were set for each sample.

### 4.7. Weighted Gene Co-Expression Network Analysis for the Construction of Modules

We performed a weighted gene co-expression network analysis (WGCNA) on normalized RNA-Seq data by an R-package [72]. All low-expression and low-coefficient-of-variation genes were removed (FPKM < 1 and CV < 0.1). The soft thresholding value was 12 and the gene cluster dendrogram was performed with a height cutoff of 0.25, based on a recommendation from the WGCNA tutorial [73]. The modules were obtained by the automatic network construction function, with default parameters in the WGCNA software package (version 1.63). The correlations between modules and traits were calculated by the Pearson method. The top 30 genes including the candidate hub genes network were visualized by Cytoscape (version 3.9.1). Each of the top 10 genes with the maximum for each module’s connectivity was considered a “highly connected gene” (hub gene) [74]. The module genes were used as a query for the prediction of the potential TFs (transcription factors) by the PlantTFDB database (http://planttfdb.gaolab.org/, accessed on 11 October 2019).

### 4.8. Statistical Analysis

One-way analysis of variance (ANOVA) was performed using SPSS 26.0 (IBM, Armonk, NY, USA) to compare mean values, and differences were compared using LSD (least significant difference) multiple comparison tests with a significance level of *p* < 0.05 and Student’s *t*-test. Pearson correlation coefficient (r) analysis was used to determine the correlation coefficients between traits. GraphPad Prism version 9 (GraphPad Software, Inc., San Diego, CA, USA) was used for data visualization.

## Figures and Tables

**Figure 1 ijms-25-01648-f001:**
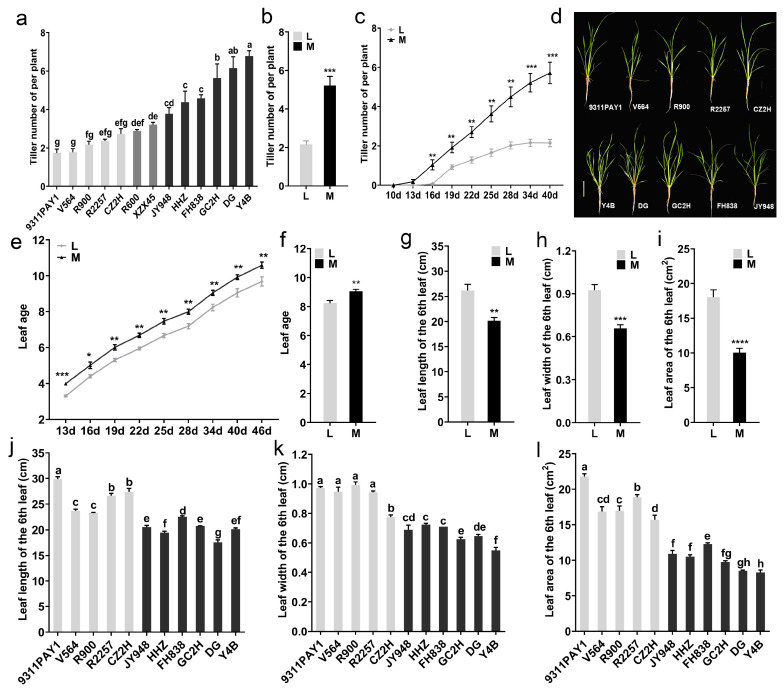
Rice tillering and related phenotypic traits. (**a**) The tiller number of per plant of the 13 rice varieties at 34 days after sowing (*n* = 3, each replication contained 10 plants). (**b**) The average values of tiller number per plant of the two rice groups at 34 days after sowing. (**c**) Dynamic changes of the tiller number per plant of the two rice groups. (**d**) Plant morphology of each variety at 28 days after sowing. Bar = 10 cm. (**e**) Dynamic changes of leaf age of the two rice groups. (**f**) The average values of leaf age of the two rice groups at 34 days after sowing. (**g**–**i**) The average values of leaf length (**g**), leaf width (**h**) and leaf area (**i**) of the 6th leaf of the two rice groups. (**j**–**l**) The length (**j**), width (**k**) and leaf area (**l**) of the 6th leaf of the 11 rice varieties (*n* = 3, each replication contained 10 plants). Note: L: Low-tillering group, M: Multi-tillering group. The values are mean ± SE. The significance of the difference between the low-tillering group and the multi-tillering group was determined using Student’s *t*-test (* *p* < 0.05, ** *p* < 0.01, *** *p* < 0.001, **** *p* < 0.0001). The different lowercase letters in (**a**,**j**–**l**) indicate significant differences according to LSD (least significant difference) tests (*p* < 0.05).

**Figure 2 ijms-25-01648-f002:**
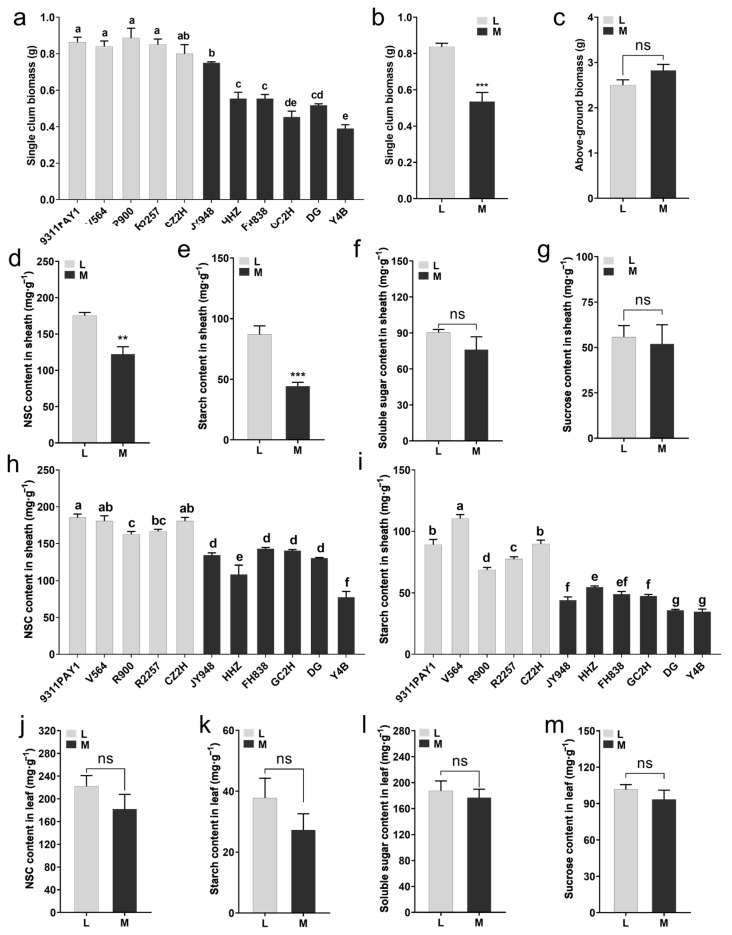
Aboveground biomass at 28 days after sowing and the non-structural carbohydrate (NSC) content of leaves and sheaths at 6-leaf stage. (**a**) The single-culm biomass of the 11 rice varieties at 28 days after sowing (*n* = 3, each replication contained 10 plants). (**b**,**c**) The average values of single-culm biomass (**b**) and above-ground biomass (**c**) of the two rice groups at 28 days after sowing. (**d**–**g**) The average contents of NSC (**d**), starch (**e**), soluble sugar (**f**) and sucrose (**g**) in sheaths of the two rice groups. (**h**,**i**) The contents of NSC (**h**) and starch (**i**) in sheaths of the 11 rice varieties (*n* = 3, each replication contained 10 plants). (**j**–**m**) The average contents of NSC (**j**), starch (**k**), soluble sugar (**l**) and sucrose (**m**) in the leaves of the two rice groups. Note: L: Five low-tillering rice varieties, M: Six multi-tillering rice varieties. Note: L: Low-tillering group; M: Multi-tillering group. Values are mean ± SE. The significance of the difference between low-tillering group and multi-tillering group was determined using Student’s *t*-test (** *p* < 0.01, *** *p* < 0.001, “ns” represents no significant difference). The different lowercase letters in (**a**,**h**,**i**) indicate significant differences based on the LSD (least significant difference) test (*p* < 0.05).

**Figure 3 ijms-25-01648-f003:**
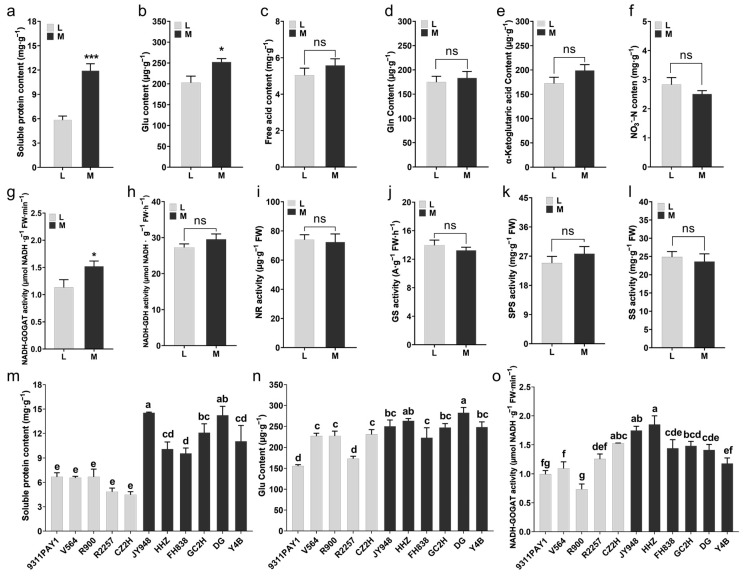
The contents of nitrogen metabolites and the activities of carbon and nitrogen metabolism-related enzymes in rice leaves at tillering stage. (**a**–**f**) The average values of soluble protein content (**a**), Glu content (**b**), free acid content (**c**), Gln content (**d**), α-Ketoglutaric acid content (**e**) and NO_3_^−^-N content in leaves of the two rice groups. (**g**–**l**) The average values of NADH-GOGAT activity (**g**), NADH-GDH activity (**h**), NR activity (**i**), GS activity (**j**), SPS activity (**k**) and SS activity (**l**) in leaves of the two rice groups. (**m**,**n**) The contents of soluble protein (**m**) and Glu (**n**) of the 11 rice lines (*n* = 3). (**o**) The activities of NADH-GOGAT in leaves of 11 rice lines (*n* = 3). Note: L: Low-tillering group, M: Multi-tillering group. Values are mean ± SE. The significance of the difference between low-tillering group and multi-tillering group was determined using Student’s *t*-test (* *p* < 0.05, *** *p* < 0.001, “ns” represents no significant difference). The different lowercase letters in (**m**–**o**) indicate significant differences according to LSD (least significant difference) tests (*p* < 0.05).

**Figure 4 ijms-25-01648-f004:**
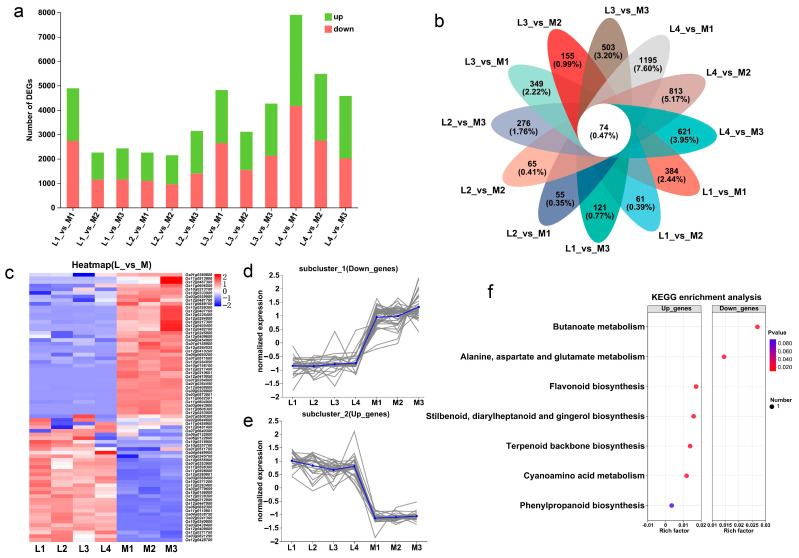
Analysis of DEGs in low-tillering rice and multi-tillering rice. (**a**) Statistics of up-regulated and down-regulated genes in different comparison pairs. (**b**) Venn analysis of core conserved DEGs in 12 comparison pairs. (**c**) Heat map of core conserved DEG expression profiles. (**d**,**e**) The expression level of the significantly down-regulated core conserved DEGs (**d**), and the significantly up-regulated core conserved DEGs (**e**). Each gray line represents the change trend of the expression level of a gene, and the blue line represents the average expression level of all genes in the subcluster. (**f**) KEGG enrichment analysis of core conserved DEGs. Note: L1: 9311PAY1, L2: V564, L3: R900, L4: R2257, M1: Yue 4B, M2: Di Gu, M3: Gui chao 2 Hao.

**Figure 5 ijms-25-01648-f005:**
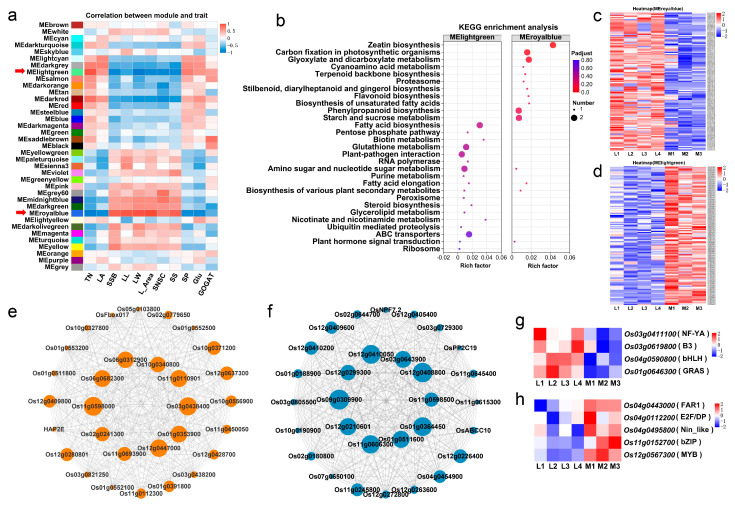
Weighted gene co-expression network analysis (WGCNA) of gene expression and the related traits. (**a**) Matrix of correlation between module and traits. TN: Tiller number per plant, LA: Leaf age, SSB: Single-stem biomass, LL: Leaf length, LW: Leaf width, L_Area: Leaf area, SNSC: Sheath NSC content, SS: Sheath starch content, SP: Soluble protein content in leaves, Glu: Glutamate content in leaves, GOGAT: Glutamate synthase activity in leaves. (**b**) KEGG functional enrichment analysis of MElightgreen and MEroyalblue module genes. (**c**,**d**) The gene expression profile heat map of the MEroyalblue module genes (**c**) and MElightgreen module genes (**d**). (**e**,**f**) The co-expressed gene network of the top 30 connectivity in the MEroyalblue module (**e**) and MElightgreen module (**f**). The size of the circle in the co-expression network diagram represents the connectivity of the gene. (**g**,**h**) The gene expression profile heat map of the transcription factors in MEroyalblue module (**g**) and MElightgreen (**h**) module genes.

## Data Availability

The raw data files of transcriptomic analysis have been uploaded to the NCBI Sequence Read Archive (SRA) database (BioProject ID: PRJNA1053911, reviewer link: https://dataview.ncbi.nlm.nih.gov/object/PRJNA1053911?reviewer=kr70dl1l8naj6hrsvso0hhc4kg, accessed on 17 December 2023). These data will be released after the paper is accepted.

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
