# Peer review of "Biochemical and Transcriptome Analyses Reveal a Stronger Capacity for Photosynthate Accumulation in Low-Tillering Rice Varieties"

_ijms, 2024, doi:10.3390/ijms25031648_

Round 1

Reviewer 1 Report

Comments and Suggestions for Authors

The author used two groups with low-tillering and multi-tillering rice varieties to reveal the mechanism of rice tillering regulation. The experiment is well designed, and also the MS is well written. But I still have some small queries to mention.

1, The introduction is too simple, the author should spend some more paragraph to well explain the C and N metabolism in regulating rice tillering, especially the previous studies by other scholars.

2, I think the quantitative real time PCR is needed to validate the transcriptome data.

In extra, some errors in the reference part should be checked carefully and should be corrected.

Reviewer 2 Report

Comments and Suggestions for Authors

The work demonstrates a commendable effort; however, some revisions are necessary before publication to enhance clarity and precision in MM section:

1. The authors should provide more detail on the specific procedures involved in seed treatment to enhance the replicability of the study.

2. Please explain the statistical methods used for data analysis to strengthen the reliability of the results. What was the rationale behind the chosen sample sizes and replication numbers?

3. Would it be possible to include information on the validation methods or quality controls employed during the biochemical analyses to enhance the accuracy and credibility of the obtained measurements?

4. Please elucidate the criteria for gene selection for validation or provide insight into the specific software utilized for RNA-Seq analysis to enhance the transparency of this section.

5. Please offer more context regarding the traits or factors used for module-trait correlations in the weighted gene co-expression network analysis. This might provide deeper insights into the genetic basis of observed traits.

6. Provide citations were necessary for introduction and MM sections

7. In my opinion it be beneficial to delve deeper into the aspects of morphology in the study.

Best Regards

Comments on the Quality of English Language

the manuscript is not presented in an intelligible fashion and is not written in standard English.

Reviewer 3 Report

Comments and Suggestions for Authors

This manuscript is written with all the required fields. I have pasted some comments for the author, which will help to improve the value of this manuscript.

> The title should be brief with some critical words. 

> Abstract is lengthy, showing the measuring information and other details.

> Keywords need revise, shouldn't belong from title or abstract part of the manuscript. 

> Introduction portion is small and should discuss more about the biochemistry part of photosynthate linked with tillering. 

> Figures if bars of different graphs presented in different color will be clearer and more understandable with respect to other axis titles. 

> Figure 4 c, d, e and f the title of some axes are very small and difficult to read.

> In Figure 5a i think there is no need to add values to the correlation boxes, which is very hard to read. If anyone wants to know the significance, it can be relate with the bar colors. 

> Last of the discussion, the conclusion paragraph should be discussed and inserted with a clear message for the reader and reviewers. 

Round 2

Reviewer 2 Report

Comments and Suggestions for Authors

.